# Discovering Photoswitchable Molecules for Drug Delivery with Large Language Models and Chemist Instruction Training

**DOI:** 10.3390/ph17101300

**Published:** 2024-09-30

**Authors:** Junjie Hu, Peng Wu, Yulin Li, Qi Li, Shiyi Wang, Yang Liu, Kun Qian, Guang Yang

**Affiliations:** 1Bioengineering Department and Imperial-X, Imperial College London, London W12 7SL, UK; j.hu@imperial.ac.uk (J.H.); qi.li120@imperial.ac.uk (Q.L.); s.wang22@imperial.ac.uk (S.W.); 2School of Chemistry and Chemical Engineering, Ningxia University, Yinchuan 750014, China; wp141@nxu.edu.cn; 3Department of Mathematics, The Chinese University of Hong Kong, Shatin, Hong Kong; liyulin@link.cuhk.edu.hk; 4Shanxi Bethune Hospital, Shanxi Academy of Medical Sciences, Third Hospital of Shanxi Medical University, Tongji Shanxi Hospital, Taiyuan 030032, China; liuyang@sxbqeh.com.cn; 5Department of Information and Intelligence Development, Zhongshan Hospital, Fudan University, 180 Fenglin Road, Shanghai 200032, China; 6National Heart and Lung Institute, Imperial College London, London SW7 2AZ, UK; 7Cardiovascular Research Centre, Royal Brompton Hospital, London SW3 6NP, UK; 8School of Biomedical Engineering & Imaging Sciences, King’s College London, London WC2R 2LS, UK

**Keywords:** drug delivery, photoresponsive molecules, quantum chemistry, language model, RLHF

## Abstract

**Background:** As large language models continue to expand in size and diversity, their substantial potential and the relevance of their applications are increasingly being acknowledged. The rapid advancement of these models also holds profound implications for the long-term design of stimulus-responsive materials used in drug delivery. **Methods:** The large model used Hugging Face’s Transformers package with BigBird, Gemma, and GPT NeoX architectures. Pre-training used the PubChem dataset, and fine-tuning used QM7b. Chemist instruction training was based on Direct Preference Optimization. Drug Likeness, Synthetic Accessibility, and PageRank Scores were used to filter molecules. All computational chemistry simulations were performed using ORCA and Time-Dependent Density-Functional Theory. **Results:** To optimize large models for extensive dataset processing and comprehensive learning akin to a chemist’s intuition, the integration of deeper chemical insights is imperative. Our study initially compared the performance of BigBird, Gemma, GPT NeoX, and others, specifically focusing on the design of photoresponsive drug delivery molecules. We gathered excitation energy data through computational chemistry tools and further investigated light-driven isomerization reactions as a critical mechanism in drug delivery. Additionally, we explored the effectiveness of incorporating human feedback into reinforcement learning to imbue large models with chemical intuition, enhancing their understanding of relationships involving -N=N- groups in the photoisomerization transitions of photoresponsive molecules. **Conclusions:** We implemented an efficient design process based on structural knowledge and data, driven by large language model technology, to obtain a candidate dataset of specific photoswitchable molecules. However, the lack of specialized domain datasets remains a challenge for maximizing model performance.

## 1. Introduction

Biocompatible materials sensitive to external physicochemical stimuli can be used for drug delivery systems [1,2,3]. Certain compounds have the ability to absorb light at specific wavelengths and subsequently modify their molecular structure, such as through conjugation, conformational changes, or isomerization [3,4,5,6,7]. The concept of light-responsiveness is gaining prominence due to the potential for creating materials that react to harmless electromagnetic radiation, particularly in the UV, visible, and near-infrared spectra [8,9]. Specifically, molecules that undergo isomerization upon excitation hold promise as molecular switches for light-responsive drug delivery [10,11]. Ultraviolet or blue light can serve as a trigger for topical treatments on the skin or mucous membranes. Near-infrared light in the wavelength range of 650 to 900 nm offers a broader range of applications. The 1931 proposal by Nobel Laureate Maria Goeppert-Mayer on two-photon absorption is expected to significantly advance the development of contemporary near-infrared photoresponsive molecules [12]. Aromatic azo compounds with the -N=N- group are a class of molecules that can undergo photoresponsive isomerization [12,13].

The rapid progress of large language modeling has spurred extensive discussions among experts regarding its effectiveness in achieving Artificial General Intelligence (AGI) [14,15]. This conversation parallels earlier debates sparked by the success of DeepMind’s AlphaGo models, which explored the relationship between reinforcement learning and AGI [16]. As AI technology matures, scientific and biomedical research has seen a surge in productivity, with researchers increasingly acknowledging AI’s pivotal role in advancing their fields and enabling more in-depth investigations. Nonetheless, the outcomes of such research remain heavily influenced by domain-specific data. The convergence of drug delivery and research on photoresponsive molecular materials with emerging AI technologies offers significant potential and societal value, paving the way for new innovations and discoveries [17,18,19,20,21].

After the Transformer and Attention algorithms were proposed, GPT-2 further demonstrated the potential of Transformer models for content generation. Researchers have considered that one possible scenario leading to AGI is that intelligence will surge as model parameter size grows. Additionally, more efficient training of models is a key trend in model development. BigBird, Llama, GPT NeoX, and Gemma are representative models in the evolving development of GPT [22,23,24,25]. The increased ability to apply language models is also directly related to Reinforcement Learning with Human Feedback (RLHF), which is a method for training models on human-provided prompts. Reinforcement learning algorithms like Proximal Policy Optimization (PPO) incorporate the reward model as a critical component [14]. The Direct Preference Optimization (DPO) algorithm uses the language model itself as an implicit reward model and has achieved excellent training results [26]. After experimenting with the drug delivery molecule design strategy based on GPT-2 [27], we continued the development of large models by introducing a new language model and an instruction training method to improve generative performance.

In this study, we employed BigBird, Gemma, and GPT NeoX to generate light-responsive drug delivery molecules. We conducted pre-training using PubChem data and fine-tuning on the QM7b dataset containing molecular excitation energies. The generated molecule data were visualized using t-distributed stochastic neighbor embedding (t-SNE) for distribution analysis. PageRank algorithms guided the selection of molecules for refined quantum chemical simulations using the time-dependent density-functional theory (TDDFT) method. Results indicated that among the selected molecules, we collated structural features of photoresponsive molecules based on literature and chemical intuition to construct a preference dataset. To enhance chemical knowledge in our model, we implemented instruction training using the DPO algorithm and chemist feedback, increasing the number of qualifying molecular structures generated by GPT NeoX from around 132 to over 400 (Table 1). Our findings highlight the need for the further integration of comprehensive chemistry knowledge into language models for designing ideal light-responsive drug delivery molecules.

## 2. Results

### 2.1. Delivery Large Language Model for Photoresponsive Molecule Discovery

Recently, advancements in large language models (LLMs) have extended into the solving of biochemical challenges. However, the development of language models tailored for designing stimulus-responsive materials in drug delivery remains underexplored. In our previous work, we fine-tuned a pre-trained GPT-2 model on the QM7b dataset and simulated the first excitation energy of molecules using the TDDFT method. As shown in Figure 1, in this study, we further trained and fine-tuned additional large language models for the generation of photoresponsive molecules. To better explore the mechanisms of photoresponsive drug delivery, we used TDDFT to analyze the photocatalytic isomerization of molecules. We also converted the chemical knowledge related to isomerization into a preference dataset and used it for instruction training. Additionally, graph networks and the PageRank algorithm were applied for the first time in our work to recommend molecular content.

Specifically, we pre-trained and fine-tuned BigBird, Gemma, and GPT NeoX. The results of their generation can be found in Figure 2. A total of 3905 candidate molecules were produced using these LLMs, with BigBird contributing 230, Gemma 1640, and GPT NeoX 2035 molecules, respectively. In the distribution plot of Figure 2, green dots represent molecules generated by BigBird, indigo crosses denote Gemma-generated molecules, and dark blue triangles signify molecules generated by GPT NeoX. The t-SNE algorithm was used here for the visualization of over 3000 molecules, where the molecular vector descriptors were based on Morgan fingerprints and physicochemical properties.

Molecules generated by the same model tend to exhibit greater similarity, with distinct disjoint segments observed between molecules generated by Gemma and GPT NeoX. Additionally, molecules produced by Gemma or GPT NeoX often cluster near those generated by BigBird.

### 2.2. Screening Molecules with QED, SA, and PageRank Score

In the de novo molecular design process, we not only need to use generative models to obtain more content but also need to screen molecules based on specific properties to obtain recommended molecules. Before considering the photoresponsive process of drug delivery molecules, QED and SA also play a positive role in narrowing down the range of candidate molecules for drug delivery.

We conducted statistical analysis on the QED values of the three model-generated molecules and depicted the results in Figure 3a. The figure demonstrates that the overall distribution of the values is more favorable in purple and indigo compared to that in tangerine. GPT NeoX, indicated by purple, and Gemma, represented by indigo blue, demonstrate similar QED performance, with GPT NeoX slightly higher in quantity while Gemma shows slightly better average values.

By combining the content of Figure 3c and Table 2, we can obtain the chemical structures of the top 10 molecules ranked by QED and their corresponding models. Out of the ten molecules chosen based on QED scores, BigBird contributed seven and Gemma contributed three. Higher QED scores generally indicate better drug-like properties.

Similarly, we conducted an analysis of the SA results. Molecules with higher SA scores were found to be easier to synthesize. When combined with the distribution graph in Figure 3b, it is evident that GPT NeoX significantly outperforms both BigBird and Gemma in this regard. Among the top ten ranked molecules in Figure 3d and Table 2, eight were contributed by GPT NeoX and two by BigBird.

To further illustrate the relationships among these molecules, we employed a knowledge graph approach. This method involves representing each molecule as a vector and assessing the similarity between any pair of molecules to generate the adjacency matrix for the knowledge graph network. The edge weights between nodes range from −1 to 1, with 1 indicating the highest similarity and −1 the lowest. The resulting molecule associations yield over 10 million matrix elements, making direct data interpretation challenging. Instead, we applied the graph network-based Page-Rank algorithm to rank nodes based on their associations comprehensively. In Figure 4a, we visualize the edge matrix of the graph network using a heatmap. In Figure 4b, we present the top 20 molecules ranked in the PageRank analysis.

### 2.3. First Excitation Energy and Photo-Isomerisation Mechanisms

In addressing the requirements for physicochemical properties and synthesis of photoresponsive molecules for drug delivery systems, we have incorporated QED Score and SA Score to prioritize all generated molecules. In order to better reflect the application potential of molecules generated from CLMs, we calculated the first excitation energies of molecules in gas ghase, water and organic solvent environments using TDDFT.

In Table 2, we present detailed computational chemical analysis and excitation energy calculations for the top 20 molecules ranked by PageRank and the top 10 molecules based on QED and SA scores. Out of all the recommended molecules, one molecule can be excited by visible light, while the excitation energies of the other molecules correspond to wavelengths in the UV spectrum. In particular, GPT NeoX-generated molecules like NC1=CSN=C=C1 exhibit excitation wavelengths near the near-infrared spectrum. This is the fifth-ranked molecule based on SA values. These molecules demonstrate an increased depth of transmission compared to others, suggesting greater potential for practical applications.

To delve deeper into the mechanisms underlying photoresponsive reactions, we conducted calculations on the photo-isomerisation of an azo compound in water, as azobenzene is a well-known molecule that exhibits photo-isomerisation between its cis and trans conformations. As illustrated in Figure 5, the azo compound (referred to as A) undergoes isomerisation between cis and trans conformations through rotation about the N=N bond.

Ground-state optimized geometries reveal that the cis isomer is energetically higher by 0.56 eV compared to the trans isomer. Additionally, TDDFT calculations yield transition energies of 4.05 eV and 4.56 eV for the S1 ← S0 transition in the cis and trans conformations, respectively. The calculated transition energy is 6 eV along the isomerization, indicating that the S2 state can be ruled out in the reaction.

The potential energy surface calculated for A is depicted in Figure 5. In the ground state, the barrier height for the cis to trans transition is measured at 1.32 eV, closely matching the highest points on the energy surface along this pathway (1.35 eV relative to the cis conformation). Conversely, in the first excited state, there exists essentially no energy barrier along the rotation pathway. Furthermore, no curve crossing between the S0 and S1 states is observed, suggesting that photo-isomerisation is less likely to occur irrespective of the excitation wavelength.

### 2.4. Instruction Training

Scientific discoveries and technological advances evolve through iterative trial-and-error processes rooted in scientific hypotheses and experimental validation [28]. We explored the mechanism of photocatalytic isomerisation through computational simulation experiments, beginning with the intuitive selection of generated molecules and the subsequent structural modification of one of them.

Based on the literature [12,13] and discussions related to TDDFT’s analysis of light-driven heterostructures, we recognize that the difference in energy levels (S0 and S1) directly impacts the occurrence of heterostructures. It is observed that the inclusion of the -N=N- functional group remains a viable approach to serve as a switch for light-responsive molecules on drug delivery nanoparticles.

Screening the generated molecules against this criterion yielded only a few satisfying examples. We then adopted a less stringent condition, requiring two chemically bonded N atoms adjacent to the uncyclic one. Under this relaxed criterion, we validated the transfer of chemical domain knowledge to CLMs through instruction training.

Additionally, we fine-tuned the GPT NeoX model using Preference Datasets as per the DPO algorithm, incorporating the Low Rank Adapter (LoRA) algorithm [29]. In this process, certain layers were frozen (shown in grey in Figure 6), with LoRA layers primarily focused on the Attention Mechanism module for training. Post-convergence under the DPO algorithm, the number of molecules meeting the relaxed chemical criterion for this model was 439 entries (Table 1). This data comparison validates the effectiveness of Chemist Instruction Training.

Further enhancements in molecule generation require specialized chemical knowledge, as well. Reinforcement Learning with Human Feedback (RLHF) algorithms like DPO provides a framework to incorporate diverse knowledge into language models. Our DPO-based chemist instruction training notably increased the number of molecules satisfying the criterion.

## 3. Discussion

Open-source efforts on large language models for text have laid a crucial foundation for drug delivery applications. However, developing practical light-responsive molecular materials still demands sustained interdisciplinary research. Among the three models utilized in our study, GPT NeoX demonstrates outstanding performance in generating light-responsive drug delivery molecules. We also considered more refined photocatalytic isomerisation mechanisms to assess the molecular generation effect and to provide an important reference for subsequent model design.

Furthermore, we identified the need for a recommendation algorithm aligned with drug delivery design metrics to enhance the overall intelligent design process.

Research on light-responsive drug delivery molecular materials driven by GPT technology has greatly improved the efficiency of discovering potential molecules. However, these candidates require further analysis and validation before they can be used in the preparation of nanocarriers for drug delivery systems. Subsequent steps include designing the molecular synthesis route, sample preparation, and experimental determination of the photochemical properties of molecular solvents and crystals. Deep learning and computational chemistry tools can also enhance the ability of experienced synthetic chemists to find synthesis routes. Ultimately, the molecules designed by GPT will also be experimentally validated in intelligent drug delivery systems.

The traditional trial-and-error model, combined with data- and knowledge-driven LLM technology, can significantly improve R&D efficiency, representing an enhancement of the fourth-generation paradigm for new material discovery [30].

## 4. Materials and Methods

**Large Language Models**: GPT-2 [31] represents a type of causal language model (CLM) utilized in our previous research on photoresponsive drug delivery molecules [27]. CLMs like BigBird, Gemma, and GPT NeoX exhibited superior performance and were employed in this study. CLMs predicts the next token in a sequence of tokens. Herein, the Byte-Pair Encoding Tokenizer for SMILES data was trained on the PubChem datasets [32]. We initially employed a vocabulary of 72 characters from the SMILES alphabet, resulting in a tokenizer size of 1072. For optimization, we utilized AdamW with cosine annealing for learning-rate scheduling, setting the initial learning rate to $5\times 10^{-4}$ and the final learning rate to $5\times 10^{-8}$. The BigBird, Gemma, and GPT NeoX models were all implemented using the Transformers package from Hugging Face. The configurations for these models were set with default parameters. Then pre-trained generative models were fine-tuned with the QM7b datasets, and the SMILES data were provided by Prof. Alexandre Tkatchenko’s Group [33,34].

**Instruction Training:** We utilized reinforcement learning with human feedback to refine the language models, employing the transformer reinforcement learning package. The preference datasets used here are structured as dictionaries containing prompts, chosen responses, and rejected responses. In the training process, we employed direct preference optimization (DPO) with chemical-knowledge-based feedback.

**Drug Likeness Score:** Drug likeness is a consideration when evaluating generative molecules for photoresponsive drug delivery. We utilized the quantitative estimate of drug-likeness (QED) as a metric, as introduced in the referenced work [35,36]. The QED metric yields a numerical score within the range of 0 to 1, where elevated scores correspond to an increased probability of drug likeness.

**Synthetic Accessibility Score:** The SA score [36,37], used to assess the ease of synthesizing drug-like molecules, rates molecules from 1 to 10 based on historical synthetic data and molecular complexity. Fragment contributions and a complexity penalty are derived from PubChem’s extensive molecule database, forming the foundation of this approach. Validation against estimations by expert chemists demonstrates strong agreement (r2 = 0.89). This method leverages big data to streamline and improve the synthesis evaluation process in molecular design.

**PageRank of Knowledge Graph Network:** We employed networkx to construct a Knowledge Graph Network for generating molecular information, followed by utilizing PageRank for scoring and molecular recommendation. The molecular features primarily encompass properties such as molecular fingerprints, drug-like characteristics, and structural alerts (SA). Additionally, we preprocessed these features using principal component analysis (PCA) and t-distributed stochastic neighbor embedding (TSNE) to build the adjacency matrix of the graph network.

**Computational Methods:** All calculations were conducted utilizing ORCA (version 5.0.4) [38]. Ground-state geometries were determined by employing the PBE0 functional [39] along with the basis set $6-311G^{*}$ in both the gas phase and in conjunction with the CPCM model [40,41,42] for solvation in water and chloroform. This method has previously been validated for its ability to accurately reproduce experimental findings [43]. For excited-state calculations, time-dependent density-functional theory (TDDFT) was employed using the PBE0 functional [39] and TZVP [44] basis set. The excited-state potential vertical excitation energies were found for each of the points in the ground-state potential energy surfaces. The resulting vertical excitation energies represent the excited-state potential at each point on the ground-state potential energy surfaces.

## 5. Conclusions

Our work provides a novel data-centric workflow where advanced LLMs, such as BigBird, Gemma, and GPT NeoX, along with first-principles computational simulations, were used. Our goal is to expedite the scientific discovery process of sophisticated light-responsive drug delivery molecules. Among the selected molecules, more human-friendly NIR-responsive molecules occupy a small percentage. In future work, these limitations can be addressed by collecting training data from existing studies or designing models to learn strategies to reduce excitation energy from existing data.

Our computational simulation experiments reveal differences in the structural stability of model-generated molecules compared to reported molecules, highlighting an additional limitation of the models discussed in our study. These models employ tokenizers that consider atomic and bonding features of chemical structures, limiting the inclusion of more physicochemical information in the generated results. One potential solution to address this limitation is to incorporate first-principles-based features for the essential functional groups of photoresponsive drug delivery molecules and to develop new language models with innovative attention mechanisms inspired by quantum computing to align with these features.

## Figures and Tables

**Figure 1 pharmaceuticals-17-01300-f001:**
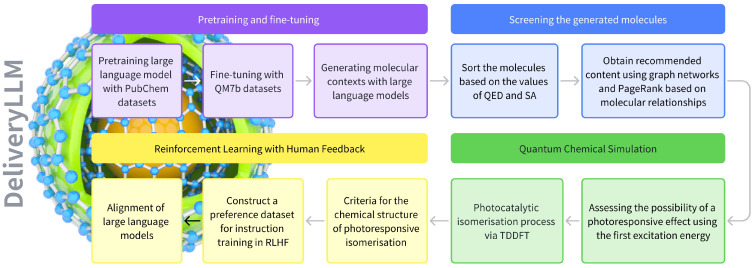
Workflow for large language models on photoresponsive isomer molecules: Pretraining and fine-tuning of the large language model, screening of generated content, quantum chemical simulation of molecular properties and mechanisms, and reinforcement learning with human feedback.

**Figure 2 pharmaceuticals-17-01300-f002:**
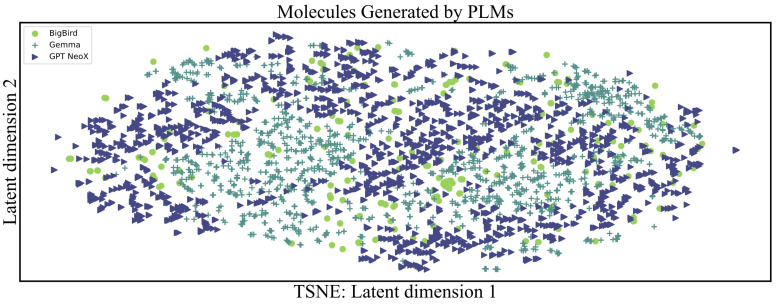
The molecular content generated by the Pre-trained Language Models (PLMs). Its visualization is based on T-SNE. The PLMs used here include BigBird, Gemma, and GPT NeoX, represented by green dots for BigBird, indigo crosses for Gemma, and dark blue triangles for GPT NeoX.

**Figure 3 pharmaceuticals-17-01300-f003:**
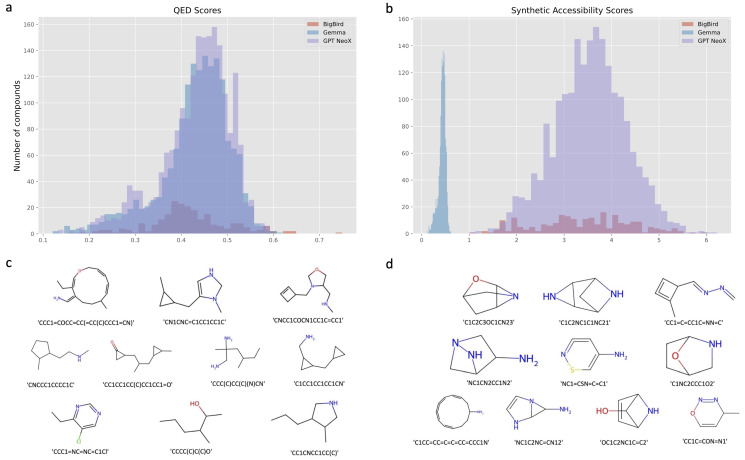
The evaluation of generative molecules. (**a**,**b**) The QED and SA scores of generative molecules for BigBird, Gemma, and GPT NeoX, respectively. Here, the data for BigBird are represented in orange-red, the data for Gemma are represented in indigo blue, and the data for GPT NeoX are represented in purple. (**c**,**d**) The chemical structures of the molecules ranked by SA and QED scores, respectively.

**Figure 4 pharmaceuticals-17-01300-f004:**
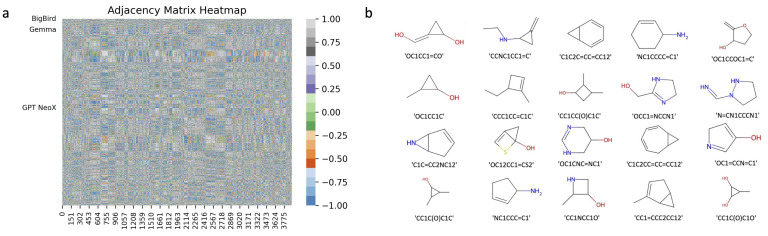
Molecule recommendation based on PageRank. (**a**) Adjacency matrix of molecular features, which is also used to implement knowledge graph networks of PageRank. (**b**) The chemical structures of the top 20 molecules ranked by PageRank score.

**Figure 5 pharmaceuticals-17-01300-f005:**
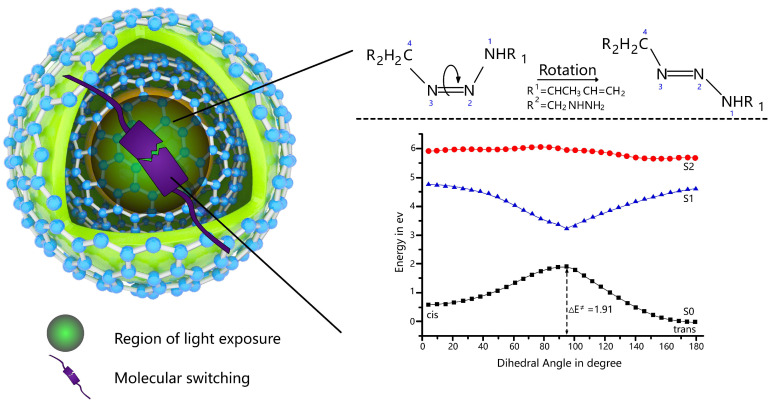
Potential energy diagrams along the isomerisation between cis and trans conformations via the rotation of dihedral (N1-N2-N3-C4) on S0, S1, and S2 states in the solvation of water. Characters 1, 2, 3, and 4 represent the four vertices of a dihedral angle. In the photocatalytic isomerization process of the molecule shown in this Figure, the initial state (cis), transition state (ΔE), and final state (trans) are illustrated. The geometric coordinate data corresponding to these states have been added to Appendix A.

**Figure 6 pharmaceuticals-17-01300-f006:**
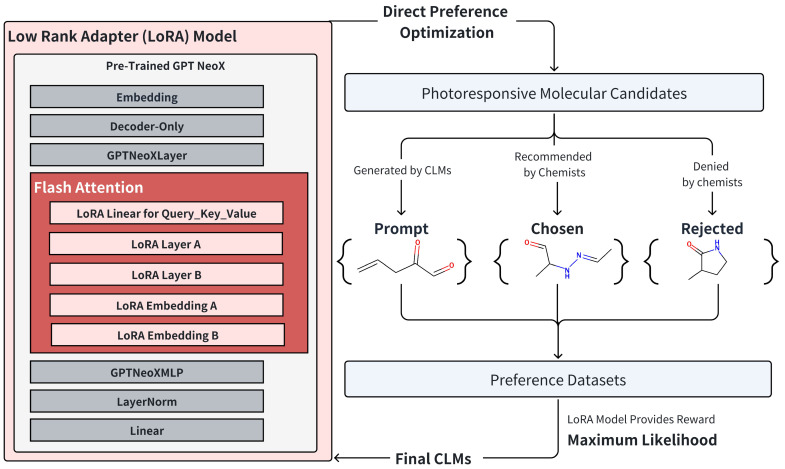
The workflow of per-trained GPT NeoX with DPO trainer.

**Table 1 pharmaceuticals-17-01300-t001:** Number of molecules meeting the chemical requirements, containing two N atoms that are chemically bonded next to each other in an acyclic form. Here, we transformed the structure containing the -N=N- functional group into one that includes nitrogen atoms that are bonded but are not part of a polycyclic ring.

Models	Number of Molecules Meeting the Chemical Requirements
BigBird	17
Gemma	79
GPT Neox	132
Chemist Instruction Training	439

**Table 2 pharmaceuticals-17-01300-t002:** The details regarding top recommendation molecules. The organic solvent used here is chlorobenzene.

Methods	SMILES	First Excitation Energy (eV)			Language Model
		Gas Phase	Water	Organic Solvents	
**PageRanks**	OC1CC1=CO	5.923	6.444	6.4	BigBird
	CCNC1CC1=C	5.066	5.142	5.121	GPT NeoX
	C1C2C=CC=CC12	4.346	4.275	4.281	GPT NeoX
	NC1CCCC=C1	5.873	6.081	6.03	GPT NeoX
	OC1CCOC1=C	6.448	6.525	6.477	GPT NeoX
	OC1CC1C	7.118	7.426	7.364	GPT NeoX
	CCC1CC=C1C	6.942	7.022	6.987	Gemma
	OCC1=NCCN1	6.945	7.372	7.277	GPT NeoX
	OCC1=NCCN1	5.922	6.16	6.113	Gemma
	N=CN1CCCN1	6.112	6.419	6.386	GPT NeoX
	C1C=CC2NC12	5.768	5.852	5.831	GPT NeoX
	OC12CC1=CS2	3.817	3.6	3.632	Gemma
	OC1CNC=NC1	5.965	6.183	6.13	GPT NeoX
	C1C2CC=CC=CC12	4.910	4.801	4.795	GPT NeoX
	OC1=CCN=C1	5.097	5.251	5.207	Gemma
	CC1C(O)C1C	7.019	7.394	7.313	GPT NeoX
	NC1CCC=C1	5.949	6.202	6.165	GPT NeoX
	CC1NCC1O	6.310	6.871	6.753	Gemma
	CC1=CCC2CC12	6.612	6.493	6.473	GPT NeoX
	CC1C(O)C1O	6.648	6.909	6.853	BigBird
**QED**	CCC1=COCC=CC(=CC(C)CCC1=CN)	3.927	3.847	3.862	BigBird
	CN1CNC=C1CC1CC1C	4.807	5.099	5.016	BigBird
	CNCC1COCN1CC1C=CC1	5.883	6.008	5.99	BigBird
	CNCCC1CCCC1C	6.055	6.319	6.255	BigBird
	CC1CC1CC(C)CC1CC1=O	3.783	3.848	3.83	BigBird
	CCC(C)CC(C)(N)CN	6.240	6.616	6.521	Gemma
	C1CC1CC1CC1CN	6.493	6.918	6.812	BigBird
	CCC1=NC=NC=C1Cl	4.516	4.639	4.606	Gemma
	CCCC(C)C(C)O	7.084	7.485	7.394	Gemma
	CC1CNCC1CC(C)	6.239	6.575	6.484	BigBird
**SA**	C1C2C3OC1CN23	7.331	7.704	7.62	GPT NeoX
	C1C2NC1C1NC21	6.555	6.969	6.869	GPT NeoX
	CC1=C=CC1C=NN=C	3.604	3.783	3.731	BigBird
	NC1CN2CC1N2	6.317	6.994	6.854	GPT NeoX
	NC1=CSN=C=C1	2.173	2.091	2.041	GPT NeoX
	C1NC2CCC1O2	6.036	6.530	6.416	GPT NeoX
	C1CC=CC=C=C=CC=CCC1N	3.613	3.481	3.465	BigBird
	NC1C2NC=CN12	5.258	5.391	5.347	GPT NeoX
	OC1C2NC1C=C2	5.146	5.45	5.365	GPT NeoX
	CC1C=CON=N1	3.501	3.594	3.568	GPT NeoX

## Data Availability

All the files about LLMs and RLHF can be obtained via our link (accessed on 26 September 2024): https://github.com/jhu22/Pharmaceuticals2024. The data and codes also can be obtained from the author by email.

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
