# Peer review of "Discovering Photoswitchable Molecules for Drug Delivery with Large Language Models and Chemist Instruction Training"

_pharmaceuticals, 2024, doi:10.3390/ph17101300_

Round 1

Reviewer 1 Report

Comments and Suggestions for Authors

I will have three major points. 

1) You should have only the 2D representation of molecules you discuss. It is not necessary to give the letter structures in the figures. This is only more confusing. The 2D representations are sufficient and give the necessary structures.

2) Coloring and figure details are not very visible. Fonts are too small. Colors are hard to distinguish throughout the figures. I cannot make sense from the figures.

3) I am computational chemist. I can tell you that having pure computational papers have dubious results. There needs to be collaboration with research scientists who do experiments. Pure theory papers can give misleading and false results. For this reason I will reject the paper until there is collaboration with experiments.

Comments on the Quality of English Language

No major issues with English.

Author Response

Comments-1: 1) You should have only the 2D representation of molecules you discuss. It is not necessary to give the letter structures in the figures. This is only more confusing. The 2D representations are sufficient and give the necessary structures.

Response-1: We appreciate the reviewer’s suggestions to make our work more concise. We partially agree with the reviewer’s opinion that retaining only the 2D representation of the molecules could make the layout more concise, especially for researchers familiar with the relationship between the 2D representation and SMILES linear notation of the molecules. However, removing the SMILES representation could cause confusion for many researchers in understanding the connection between the data in Table 1 and the 2D representations of the molecules. For these reasons, we believe that retaining the current approach will be more convenient for a broader range of researchers.

Comments-2: Coloring and figure details are not very visible. Fonts are too small. Colors are hard to distinguish throughout the figures. I cannot make sense from the figures.

Response-2: We sincerely apologize for the negative experience the reviewer had while reading our paper. These difficulties are primarily due to the large volume of data we processed. In Figure 3a, we visualized over 10 million matrix elements, which indeed makes reading the visualizations challenging. However, we chose a data processing method widely used in scientific publications. If the reviewer has any suggestions for improving the presentation of large data sets, we are eager to learn from them. For now, we have increased the resolution of all images to ensure they remain clear when zoomed in. Before making further revisions and discussions, we would appreciate receiving more specific and guiding suggestions from the reviewer.

Comments-3: I am computational chemist. I can tell you that having pure computational papers have dubious results. There needs to be collaboration with research scientists who do experiments. Pure theory papers can give misleading and false results. For this reason I will reject the paper until there is collaboration with experiments.

Response-3: We have also discussed our work with experts in drug delivery experiments, who have shown great interest in our current research. The de novo design and experimental research of photoresponsive molecules for drug delivery are closely related. The entire workflow can be divided into the following steps: de novo molecular design -> rational screening and validation -> design of synthesis pathways for candidates -> experimental preparation and testing of candidates -> synthesis of nanomaterials and subsequent experimental validation. In this workflow, introducing the latest LLMs methods in the de novo design and continuously improving their effectiveness has significant research value and societal impact. While we partially agree with the reviewer’s point that the ultimate success of photoresponsive molecules in drug delivery depends on experimental validation, it is essential to recognize that every stage in the research and development lifecycle requires technical improvements. This is where the research significance of our work lies. 

After our discussion with Prof. Lulu Cai, an expert in experimental research on drug delivery systems, we added a section discussing the relationship between 'LLM-driven design and experimental research on drug delivery.'

'''

The research on light-responsive drug delivery molecular materials driven by GPT technology has greatly improved the efficiency of discovering potential molecules. However, these candidates require further analysis and validation before they can be used in the preparation of nanocarriers for drug delivery systems. Subsequent steps include designing the molecular synthesis route, sample preparation, and experimental determination of the photochemical properties of molecular solvents and crystals. Deep learning and computational chemistry tools can also enhance the ability of experienced synthetic chemists to find synthesis routes. Ultimately, the molecules designed by GPT will also be experimentally validated in intelligent drug delivery systems. The traditional trial-and-error model, combined with data- and knowledge-driven LLM technology, can significantly improve R&D efficiency, representing an enhancement of the fourth-generation paradigm for new material discovery.

'''

Reviewer 2 Report

Comments and Suggestions for Authors

In the manuscript “Discovering Photoswitchable Molecules for Drug Delivery with Large Language Models and Chemist Instruction Training”, the authors have evaluated the use of large language models (such as GPT NeoX, BigBird and Gemma, which were refined by by reinforcement learning with human feedback) for the generation of photo-responsive drug delivery molecules, adopting a combination of computional chemistry and AI tools. Drug Likeliness score (QED metric) and Synthetic Accessibility Score (SA) were used respectively to assess the probability of drug likeliness and ease to synthesize drug-like molecules. Set of molecular information were generated by using PageRank of Knowledge Graph Network. Computional methods were adequate for the purpose (use of ORCA, ground state calculated using appropriate functional and basis set). Introduction clearly identifies the challenges and the main topic of the manuscript.

Figure 1 describes appropriately the entire scheme used in the manuscript and gives to the reader a ready and clear idea of what the authors have made and how.

The screening process of the candidate molecules (and their recommendation) are wisely described in Tables 1,2  (all the necessary parameters are present in the Table; excitation energy calculations in gas phase, water and organic solvent, regarding the 20 top molecules ranked by PageRank and the top 10 molecules based on QED and SA scores are present) and represented in Figure 3 and Figure 4. Moreover, to gain insight into the photoresponsive reactions, calculations were performed on the photoisomerization mechanism of a generic azo-compound in water (with the following trans-cis rotation) leading to quite reasonable conclusions.

Workflow of  per-trained GPT by a Direct Preference Optimization Trainer is well represented in Figure 6. Discussion is well argumented and the conclusions are consistent with data reported in the manuscript.

The approach reported in this manuscript paves the way for a deeper involvement of AI in the decisional processes behind the discovery of drug delivery molecules. Obviously, the approach has some limitations, as correctly identified by the authors, but it has a great potential too.

There are only minor issues and for these reasons my overall recommendation is “minor revisions”.

///COMMENTS///

Line 18: “can” and not “an”.

Line 199: Figure 5 instead of “Figure 4”.

Table 1: Which organic solvent has been chosen?

Figure 3a and Figure 3b: In The figures, in the upper part, “BigBrid” instead of “BigBird” is present. No title on the vertical axle (like “Number of compounds”) is present.

Figure 5: potential curve for the state S2 is present in the Figure, but nothing has been written about it.

Moreover, in the notes below the figure, molecule need to specified (conformations of which molecule?)

Optimized geometries for calculations (XYZ coordinates) should be present in a Supplementary Material file (if it is possible)

Table 2: How were performed the chemical judgements? What were the criteria?

Author Response

We greatly appreciate the reviewers' recognition of the significance of using LLM tools for the early exploration of light-responsive molecules with potential drug delivery applications. The complexity of molecular and material science is one of the driving forces behind our continuous search for new tools to enhance efficiency across different stages of research and development. When introducing new tools, we aim to validate them within existing theoretical and scientific frameworks to uncover their characteristics and limitations. Through such discussions, this work can serve as a reference for other researchers engaged in similar fields. We also extend our gratitude to the reviewers for acknowledging the clarity of our paper's approach and the importance of our research.

Comments-1:  Line 18: “can” and not “an”.  Line 199: Figure 5 instead of “Figure 4”. 

Reply-1: Thank you very much to the reviewer for pointing out our mistake. The corrections have been made as follows:

“Biocompatible materials sensitive to external physicochemical stimuli can be used for 18
drug delivery systems [ 1 –3 ]. ”

“As illustrated in Figure 5, the azo compound (referred to as A) undergoes 145 isomerisation between cis and trans conformations through rotation about the N=N bond.”

“The potential energy surface calculated for A is depicted in Figure 5.”

Comemnts-2: Table 1: Which organic solvent has been chosen?

Reply-2: Thank you very much to the reviewer for pointing out this issue. This will help improve the reproducibility of our work. The organic solvent used here is chlorobenzene. We have also added this information to Table 1.

“Table 1. The details about top recommendation molecules. The organic solvent used here is chloroben- zene.”

Comments-3: Figure 3a and Figure 3b: In The figures, in the upper part, “BigBrid” instead of “BigBird” is present. No title on the vertical axle (like “Number of compounds”) is present.

Reply-3: We have replaced 'BigBrid' with 'BigBird' in Figures 2, 3, and 4. Additionally, as per the reviewer's suggestion, we have added the title 'Number of compounds' to the vertical axis of Figures 3a and 3b.

Comments-4: Figure 5: potential curve for the state S2 is present in the Figure, but nothing has been written about it.

Reply-4: We appreciate the reviewer’s comments. The high transition energies of ~6 eV along the rotation indicates S2 state can be ruled out and thus not discussed in the present paper. We have added the following sentence in the revised manuscript, such as “The calculated transition energies is ~6 eV along the isomerization, indicating S2 state can be ruled out in the reaction.”

Comments-5: Moreover, in the notes below the figure, molecule need to specified (conformations of which molecule?) Optimized geometries for calculations (XYZ coordinates) should be present in a Supplementary Material file (if it is possible)

Reply-5: We appreciate the reviewer’s comments. We have added the optimized cartesian coordinates of the key structures for A isomerization in the appendix. At the same time, we have added supplementary explanations in the figure caption.

"In the photocatalytic isomerization process of the molecule shown in the figure, the initial state (cis), transi- tion state (∆E), and final state (trans) are illustrated. The geometric coordinate data corresponding to these states have been added to the appendix."

Optimized coordinates for A isomerization through rotation about the N=N bond in water.

Cis

C        3.414338000      1.148000000     -2.948062000

C        2.453240000      1.810094000     -2.304456000

C        1.030152000      1.340845000     -2.171861000

N        0.565434000      1.436787000     -0.765514000

N       -0.018235000      0.420347000     -0.057741000

N        0.468095000     -0.721198000      0.045662000

C        1.764858000     -1.106221000     -0.513332000

C        1.600521000     -2.202974000     -1.562107000

N        2.895426000     -2.782005000     -1.893707000

N        2.820639000     -3.776054000     -2.909832000

C        0.103066000      2.185822000     -3.046965000

H        4.419490000      1.550929000     -3.043161000

H        3.226481000      0.179493000     -3.410081000

H        2.662630000      2.782510000     -1.856343000

H        0.947292000      0.298006000     -2.491054000

H       -0.052585000      2.232296000     -0.652310000

H        2.360560000     -0.269106000     -0.892697000

H        2.310142000     -1.545726000      0.332253000

H        0.966821000     -2.997542000     -1.148498000

H        1.078735000     -1.810570000     -2.455011000

H        3.498891000     -2.049785000     -2.260349000

H        2.452002000     -4.616168000     -2.468425000

H        2.122588000     -3.506686000     -3.610575000

H       -0.930944000      1.833977000     -2.962802000

H        0.412339000      2.123030000     -4.094814000

H        0.140158000      3.239718000     -2.744598000

trans

C       -3.646803000      1.358683000      1.418816000

C       -2.518253000      2.060561000      1.331076000

C       -1.153511000      1.451954000      1.183465000

N       -0.573440000      1.914885000     -0.079189000

N        0.312755000      1.210229000     -0.777589000

N        0.844090000      0.253945000     -0.162598000

C        1.805638000     -0.449724000     -0.996872000

C        1.360325000     -1.895054000     -1.192361000

N        2.379562000     -2.647597000     -1.910045000

N        1.995615000     -3.989780000     -2.189774000

C       -0.250743000      1.816593000      2.364112000

H       -4.610300000      1.845410000      1.548604000

H       -3.646050000      0.271044000      1.371167000

H       -2.547843000      3.150532000      1.387654000

H       -1.238777000      0.357561000      1.121782000

H       -1.146120000      2.488188000     -0.684023000

H        1.928427000      0.050917000     -1.968502000

H        2.775745000     -0.448599000     -0.482651000

H        1.213669000     -2.366055000     -0.211765000

H        0.382437000     -1.911181000     -1.709156000

H        2.535479000     -2.207066000     -2.813728000

H        2.099137000     -4.509156000     -1.320241000

H        0.995211000     -4.025956000     -2.410894000

H        0.751887000      1.405866000      2.216206000

H       -0.664234000      1.409150000      3.292863000

H       -0.173773000      2.904888000      2.465215000

Transition state

C       -4.213817000     -0.450788000      0.968925000

C       -3.557760000      0.108145000     -0.042077000

C       -2.086138000      0.393378000     -0.029112000

N       -1.436466000     -0.430277000     -1.049211000

N       -0.340585000     -1.129848000     -0.901860000

N        0.214890000     -1.130837000      0.246591000

C        1.252794000     -0.276054000      0.733660000

C        2.579703000     -0.424941000     -0.005454000

N        3.581024000      0.471451000      0.554781000

N        4.815804000      0.439462000     -0.159520000

C       -1.793535000      1.872716000     -0.249617000

H       -5.285333000     -0.623124000      0.923392000

H       -3.704543000     -0.750367000      1.882011000

H       -4.093875000      0.405769000     -0.944011000

H       -1.639253000      0.059269000      0.912941000

H       -1.924269000     -0.616816000     -1.916507000

H        0.950461000      0.787712000      0.705283000

H        1.411056000     -0.515103000      1.788259000

H        2.937138000     -1.454512000      0.097691000

H        2.423677000     -0.240723000     -1.082872000

H        3.236228000      1.421063000      0.467384000

H        5.300848000     -0.402504000      0.131311000

H        4.627335000      0.314620000     -1.155589000

H       -0.717104000      2.059337000     -0.260440000

H       -2.237603000      2.462970000      0.556141000

H       -2.214929000      2.218022000     -1.198421000

Comments-6: Table 2: How were performed the chemical judgements? What were the criteria?

Reply-6: Based on previously reported literature and TDDFT calculations, compounds containing the -N=N- functional group are considered to have potential as photoresponsive molecules. We have summarized the structural features that the molecules should meet in lines xxx-xxx. Accordingly, we have counted the number of compounds generated by different LLM methods that meet these features, and the results are presented in Table 2.

We ran RLHF to train a GPT model, which improved generation performance, significantly increasing the number of molecules that meet these criteria.

The details of these standards are further clarified in the notes of Table 2. The modifications are as follows:

Number of molecules meeting the chemical judgements, which contains two N atoms that are chemically bonded next to each other in an acyclic form. Herein, we transformed the structure containing the -N=N- functional group into one that includes nitrogen atoms that are bonded but are not part of a polycyclic ring.

Reviewer 3 Report

Comments and Suggestions for Authors

The manuscript pharmaceuticals-3178542 is a contribution to the development of advanced methods for the controlled release of drugs, in this case induced by light. Both for its approach (concise, but clear) and for the methodologies, results and conclusions, it responds to the highest standards to be recommended for acceptance by Pharmaceuticals. It would also be recommended for Pharmaceutics.

The manuscript should/can be accepted in its current version.

 I can only recommend two aspects that should be taken into account when editing the final version, which have nothing to do with its content. (a) Insert the [referece(s) number(s)] before the period at the end of each sentence and preceded by a blank space. I.e. word [x-y]. (b) Adapt the style of the bibliographical references to that of the journal!!!

Author Response

We sincerely thank the reviewer for their positive feedback on our work. In future research, we will continue to improve the effectiveness of LLMs in the design of such molecular materials.

Comments: I can only recommend two aspects that should be taken into account when editing the final version, which have nothing to do with its content. (a) Insert the [referece(s) number(s)] before the period at the end of each sentence and preceded by a blank space. I.e. word [x-y]. (b) Adapt the style of the bibliographical references to that of the journal!!!

We appreciate the reviewer's reminder. We have revised all citations according to the style of the bibliographical references. The modifications can be seen in the document we have uploaded.

Reviewer 4 Report

Comments and Suggestions for Authors

Dear authors,

thank you or your well arranged research article. In my opinion the manuscript is sufficient and there are no further corrections necessary. Introduction provides sufficient information. M&M section is well arranged and all necessary issues are explained. Results are presented in a sufficient and well structured way. Recent references are included. Figures and tables are fine.

Author Response

Comments: thank you or your well arranged research article. In my opinion the manuscript is sufficient and there are no further corrections necessary. Introduction provides sufficient information. M&M section is well arranged and all necessary issues are explained. Results are presented in a sufficient and well structured way. Recent references are included. Figures and tables are fine.

Replys: We are very grateful to the reviewer for their positive feedback on our work. In the resubmitted manuscript, we have corrected formatting errors and spelling mistakes in the figures. Additionally, we have added a section to the discussion to explain the relationship between LLM-driven discovery of light-responsive molecules and experimental research on drug delivery systems. Finally, we have provided the geometric coordinate data for the initial state (cis), transition state, and final state (trans) of the molecule during the photocatalytic isomerization process shown in Figure 5 in the appendix. These additions aim to meet the interests of researchers from different backgrounds.

Round 2

Reviewer 1 Report

Comments and Suggestions for Authors

1) Fig. 2 What are latent dimensions

2) Figs. 2 and 3. The legend describes the color code. You can describe the color code in the caption but it is not necessary to describe the color code in the main text( lines 88-90 and 104-107).

Author Response

Comments 1: Fig. 2 What are latent dimensions

Reply 1: We greatly appreciate the reviewers' comments. In Figure 2, we used the TSNE visualization method to display the molecules designed by LLMs, which helps researchers intuitively observe the differences in the generated content.
According to the description of the TSNE visualization method, it maps the data into the latent space after processing. The resulting values are dimensionless, and similar terminology appears in many works that use the TSNE method and the label of “latent dimensions”, such as in the work by Zhong, M., Tran, K., Min, Y. et al. titled 'Accelerated discovery of CO2 electrocatalysts using active machine learning,' published in Nature, 581, 178–183 (2020)."

Comments 2: Figs. 2 and 3. The legend describes the color code. You can describe the color code in the caption but it is not necessary to describe the color code in the main text( lines 88-90 and 104-107).

Reply 2: 

Thank you to the reviewer for pointing out this issue. We have added a description of the label colors in the captions of Figures 2 and 3. These changes have greatly improved the readability of the images in our paper. In order to maintain the coherence of the original text, we have kept the content in the main body unchanged. The revised content is as follows:" 

Figure 2. The molecular content generated by the Pretrained Language Models (PLMs). Its visualiza- tion is based on T-SNE; The PLMs used here include BigBird, Gemma, and GPT NeoX, represented by green dots for BigBird, indigo crosses for Gemma, and dark blue triangles for GPT NeoX.
Figure 3. The evaluation of generative molecules (a) and (b) The QED and SA score of generative molecules for BigBird, Gemma, and GPT NeoX. respectively. Here, the data for BigBird is represented in orange-red, the data for Gemma is represented in indigo blue, and the data for GPT NeoX is represented in purple. (c) and (d) The chemical structures of the molecules ranked by SA and QED scores, respectively.
"
At the same time, we have marked the revised content in the attached file we uploaded.
